# GUIDED REINFORCEMENT LEARNING WITH ROLL-BACK

## ABSTRACT

Reinforcement learning-based solutions are increasingly being considered as strong alternatives to classical system controllers, despite their significant sample inefficiency when learning controller tasks from scratch. Many methods that address this issue use prior task knowledge to guide the agent's learning, with several recent algorithms providing a guide policy that is sometimes chosen to execute actions instead of the learner policy. While this approach lends excellent flexibility as it allows the guide knowledge to be provided in any format, it can be challenging to decide when and for how long to use the guide agent. Current guide policy-based approaches typically choose a static guide sampling rate empirically and do not vary it. Approaches that transfer control use simple methods like linear decay or require hyperparameter choices that strongly impact the performance. We show that under certain assumptions, the sampling rate of the guide policy can be calculated to guarantee that the mean return of the learning policy will surpass a user-defined performance degradation threshold. To the best of our knowledge, this is the first time a performance guarantee has been established for a guided RL method. We then implement a guided RL (GRL) algorithm that can make use of this sample rate and additionally introduce a roll-back feature in guided RL with roll-back (GRL-RB) to adaptively balance the trade-off between performance degradation and rapid transfer of control to the learner. Our approach is simple to implement on top of existing algorithms, robust to hyperparameter choices, and effective in warm-starting online learning.

## 1 INTRODUCTION

The sample inefficiency of model-free reinforcement learning (RL) remains one of the greatest barriers to the wider adoption of RL-trained models in everyday life (Yu, 2018). Real-world applications of RL are increasing, such as in refining the large language model ChatGPT (Ouyang et al., 2022), or for efficient cooling of commercial buildings (Luo et al., 2022). However, real-world environments often translate to complex environments for RL agents (Dulac-Arnold et al., 2021). This can be due to sparse rewards, high stochasticity and very large observation and action spaces. As such, RL agents may require extensive training to be effectively deployed in such environments.

Many systems currently run by controllers or human operators demonstrate potential performance gains from automation. Often in such systems, there is an available policy that governs the behaviour of the original controller (e.g., a set of fuzzy or crisp rules). Alternatively, in a system controlled by a human operator, data that describes the behaviour running the system could be collected. Rather than forcing an RL-based controller to commence learning from scratch, such prior policies or expert data could be exploited to warm-start an RL agent before deployment in an online environment (Luo et al., 2022). Additional challenges of real-world environments make a warm-starting solution more attractive. Firstly, while the sample inefficiency of model-free RL is often handled by training across many parallel environments (Dulac-Arnold et al., 2021), real-world environments cannot be similarly replicated. Further, for many applications, there is a need to ensure a certain level of performance when a model is deployed to the real world, e.g. for robotics (Zhao et al., 2020), (Lobbezoo et al., 2021) and self-driving cars (Maramotti et al., 2022)(Isele et al., 2018). These types of agents would ideally commence at a reasonable performance level, such that they require only fine-tuning in the online environment. It is important that a warm-starting solution can not only accelerate the learning of the agent but minimise performance degradation after transfer to the online environment.

In this paper, we focus on a 'roll-in' method, where control is gradually shifted from a guide policy to a learning policy in a stepped manner. The focus of our Guided Reinforcement Learning (GRL) method is to allow online learning while reducing performance degradation. We show that given a user-defined acceptable degradation threshold, the online agent can be theoretically guaranteed to maintain an evaluation return above this threshold under certain assumptions. The most challenging of these assumptions is that the agent is able to converge fully between roll-in steps. To ease the challenges in meeting this assumption, guided RL with roll-back (GRL-RB) is introduced to enable tuning of the guide sampling rate, to respect the given threshold. Further, GRL-RB is flexible in the structure of the prior policy (e.g. heuristics, decision trees, a policy learned through imitation learning/offline RL etc.) and can be implemented on top of existing algorithms. Finally, this method gradually transfers full control to the learning agent, to ensure exploration is not hindered by the prior knowledge and to minimise distribution mismatch between the replay buffer trajectories and the trajectories induced by the learning agent's policy (Fujimoto et al., 2019).

Our contributions are threefold: a) A guided reinforcement learning approach (GRL) that enables the gradual transfer of control to a learning policy, with a guaranteed online performance above a user-defined threshold. (b) The theoretical derivation and experimental confirmation of guide sampling rates for GRL under defined assumptions. (c) A GRL with a roll-back algorithm (GRL-RB) that helps to retain the performance guarantee of GRL while relaxing its assumptions.

## 2 PRELIMINARIES AND RELATED WORK

**Markov Decision Process (MDP).** An RL problem must be representable as a MDP (Bellman, 1957). Such a problem can be described by a 5-tuple $\langle S, A, P, R, \gamma \rangle$, where $S$ is the state space, $A$ is the action space, $P : S \times A \times S \to [0, 1]$ describes the next state transition probability $P(s_{t+1}|s_t, a_t)$ given the current state and action, $R$ describes the reward function $S \times A \to \mathbb{R}$ and $\gamma \in [0, 1]$ is a discount factor. A trajectory $\tau$ in an MDP describes a finite sequence of states and actions:$(s_0, a_0, s_1, a_1, ..., s_{T-1}) \in S \times A$ taken in the environment.

**Policy.** A policy $\pi$ governs the next action $a_{t+1}$ an agent will take in an MDP - depending on the current state $s_t$ - resulting in a step reward $r_t$. The policy can be stochastic $a_{t+1} \sim \pi(.|s_t)$ or deterministic $a_{t+1} = \mu(s_t)$.

**Guide, Learning and Online Sampling Policies.** The guide and (online) learner policies will be denoted $\pi_g$ and $\pi_l$ respectively. The online sampling policy $\pi$ is a combination of $\pi_g$ and $\pi_l$, depending on which policy is chosen to be sampled at $t$.

**Undiscounted Episodic Return.** For a discrete action and observation space, the undiscounted episodic return $\hat{R}_\pi$ for an agent in an MDP following $\pi$ is:

$$\hat{R}_\pi = \sum_\tau \left( \left[ \rho_0(s_0) \prod_{t=0}^{T-1} p(s_{t+1}|s_t, a_t)\pi(a_t|s_t) \right] \sum_{t=0}^{T-1} r_t \right) \tag{1}$$

where $\rho_0$ is the initial state distribution (Achiam, 2018).

One way to categorize warm-start algorithms is by focusing on the area of the RL pipeline where knowledge is inserted, namely, replay buffer, parameter updates, initial function parameters and action selection. We review action selection-based methods in this section, as they are the most relevant to our work, and include a review of the other methods in Appendix A. We also review the roll-back concept integral to our GRL-RB algorithm, and its previous use in the literature.

### 2.1 ACTION SELECTION

Action selection-based methods use either an online learning agent initialised with randomised parameters or a guide policy to choose actions. This method has become increasingly popular due to its inherent flexibility - the guide policy can be of any format - e.g. a pre-trained RL policy, a decision tree, a heuristic or a list of rules (Chang et al., 2015; 2023; Liu et al.; Uchendu et al., 2023). An obvious challenge in any guide policy-based approach is the question of how and when to reduce or remove the influence of the guide agent during the learner's training. This is especially important when the underlying algorithm is on-policy, such that the data distribution eventually reflects the distribution induced by the learner's policy. We find that most current approaches simply choose the

sampling rate empirically, and often do not vary it throughout training. Such a method for supervised learning was used in Daumé et al. (2009), where subsequent 'guide' classifiers are created through linear interpolation $\beta h' + (1 - \beta)h$ of the previous guide $h'$ and current policy $h$. While the choice of the constant $\beta$ is discussed in terms of providing a bound on the supervised loss, it is not varied during training. Another linear interpolation uses a constant $\beta$ to interpolate between a continuous action $u^{MPC}(t)$ from a Model Predictive Control method, and a DDPG policy action $a(t)$.

A number of methods used horizon-based sampling (Chang et al., 2015; 2023; Liu et al.; Uchendu et al., 2023; Daoudi et al., 2024; Yang et al., 2022; Zhang et al., 2024) where a guide policy is sampled until some horizon $h$ is reached, after which the learner probabilistically chooses the actions (or vice versa). Alternatively, each time step may have a probability of sampling the learner. These probabilities may not vary throughout training (Chang et al., 2015; 2023), or may be gradually reduced. In Liu et al., prior to $h$ (sampled randomly), one of many oracles or the learner itself is sampled, depending on a prediction of the oracle's expertise in that state by their learned value functions. As the learner improves, it will tend to be chosen over the oracles. In Jump-Start Reinforcement Learning (JSRL) proposed by Uchendu et al. (2023), sampling the guide first allows the learner agent to start closer to its goal near the episode end, giving the agent an easier task to reduce performance degradation. While this means that the learner should initially be sampled for short periods that gradually increase, in practice this depends highly on having constant $N$. This is because if the episode runs for longer than $N$, the learner will be sampled. In some environments, this may leave the agent unable to reach the goal within the environment's time limit $T$. If $T >> N$, the learner may end up being sampled for most of the episode, as shown in Appendix B.

Daoudi et al. (2024) instead used multiple local guides. The learner policy is used for action selection until a local guide satisfies a confidence function indicating their knowledge of the current state. The guide then produces an action which is then modified by a bounded, parameterised perturbation. This was an effective method for continuous action spaces, though the guide is again used throughout learning. In Yang et al. (2022), control in a safety-focused constrained MDP is handed back to the guide when a safety violation occurs. Again, no procedure is implemented to ensure control is fully transferred to the learner. While a linear decay method is implemented, it was not effective compared to the safety-violation based method. Finally, an adaptive sampling approach in (Zhang et al., 2024) produced samples for an imitation-learning based policy by framing sampling as a MDP, with the action space involving either sampling the online learner or expert. However, given the sampler begins as a randomised policy, it would be difficult to make the performance guarantees we are interested in using this method. This is similarly true for Liu et al., where performance would be very difficult to guarantee while the oracles' value functions are being learned.

## 2.2 ROLL-BACK

The concept of reverting to safe or previous operational states enhances the safety and stability of reinforcement learning systems. (Hans et al., 2008) introduced an approach where safe actions are identified based on a modified Bellman equation that prioritizes minimal future rewards rather than maximal returns. This strategy involves maintaining a backup policy to revert to when the current policy encounters unsafe or suboptimal performance. Actions leading to "fatal" transitions—where rewards fall below a threshold—are avoided, and the agent reverts to previously validated safe actions using a level-based exploration strategy. (Ma et al., 2019) expanded the "roll back" principle to a novel regret-based mechanism within a navigation agent. This mechanism enhances the navigation strategy by allowing the agent to revert to better alternatives based on past decisions. This retrospective analysis is supported by heuristic aids and progress estimation, enabling the agent to continuously refine its decision-making process. The recovery mechanism in Dasagi et al. (2019) allows the RL agent to revert to previous, stable versions of its policy upon detecting performance drops using statistical methods like the Mann-Whitney U-test. This recovery strategy helps mitigate the risks associated with aggressive policy updates that can degrade learning progress, ensuring more reliable and continuous improvement over time. GRL-RB relies on a similar principle to the above method, where instead the previous safe policy is the guide policy.

## 3 GUIDED REINFORCEMENT LEARNING (GRL)

The first focus of this work is to enable a $\pi_l$ to explore an environment while sampling a $\pi_g$ with sampling rate $\alpha$, which is carefully chosen to guarantee that performance doesn't fall below a user-

defined performance threshold. We also wish to fully transfer control to $\pi_l$ by the end of training, while avoiding the issues with over-sampling $\pi_l$ that were discussed in Section 2. We chose to employ a similar sampling approach to Chang et al. (2015) and Chang et al. (2023), but where $\pi_g$ is sampled with probability $\alpha$ at any time during the episode. Additionally, we track the fractional use of $\pi_l$ during an episode, and only enable sampling of $\pi_l$ if $\frac{n_{\pi_l}}{t} < \alpha$, where $\frac{n_{\pi_l}}{t}$ is the fraction of time steps during which $\pi_l$ has been sampled by time step $t$. Additionally, we combine this with a curriculum approach, by gradually reducing $\alpha$ to 0 throughout training. We use a similar method to JSRL (Uchendu et al., 2023) in deciding when to move to the next curriculum stage, which occurs when the evaluation return improves upon $\hat{R}_{\pi_g}$. Our approach, Guided Reinforcement Learning (GRL) is formally described in Algorithm 1 of Appendix D. To enable the implementation of GRL, we derive its $\pi_g$ learning rate $\alpha$ below.

### 3.1 DERIVING GUIDE SAMPLING RATES FOR GRL

To ensure that the agent evaluation score remains above a user-defined performance degradation threshold, we derive a sampling rate $\alpha$ for $\pi_g$ to be used in GRL. The *evaluation score* refers to the mean undiscounted return for an episode in the environment. Initially, the guide policy $\pi_g$ is evaluated for $N_i$ episodes (prior to commencing online training), to determine its mean evaluation score $\hat{R}_{\pi_g} = \frac{1}{N_i} \sum_{n=1}^{N_i} R_n$. The user-defined performance degradation threshold is a value between the minimum possible score and $\hat{R}_{\pi_g}$, representing the minimum mean score that the user can accept for the policy $\pi$. The user can define this by choosing $\mu \in [0, 1]$, as per Equation 2. The score for this policy is written as $\hat{R}_\pi = \frac{1}{N} \sum_{n=1}^{N} R_n$. The degradation threshold is defined as:

$$\hat{R}_\pi >= r_{min} + \mu(\hat{R}_{\pi_g} - r_{min}) \tag{2}$$

We provide the below results for the sampling rate derivations and their experimental validation.

#### 3.1.1 METHOD: DERIVATION

To simplify the derivation of $\alpha$, we modify the toy 'Combination Lock' MDP (Uchendu et al., 2023). This is an episodic MDP with fixed time horizon $H$ (corresponding to the length of the combination code). The agent must correctly choose the next digit in the combination sequence (the optimal action, $a*$), otherwise if the agent chooses the wrong digit (the non-optimal action, $\overline{a^*}$), the episode ends immediately. All rewards are 0, unless the agent reaches the end of the combination, where it receives a sparse reward of $r$ (see Appendix C for a full description of the environment). Sampling rate $\alpha$ will be derived for three variations of $\pi_l$, $\pi_g$ and the reward: variation 1 - optimal $\pi_g$, terminating $\pi_l$, sparse reward; variation 2 - non-optimal $\pi_g$, non-terminating $\pi_l$, sparse reward; and variation 3 - non-optimal $\pi_g$, non-terminating $\pi_l$, dense reward. The derived $\alpha$ is then used as input to GRL (Algorithm 1), and online training can commence. If the online algorithm is sufficient, it will allow the learner policy to converge (such that $\hat{R}_\pi = \hat{R}_{\pi_g}$). At this point, rather than always choosing non-optimal action, the learning policy $\pi_l$ now has (at least) a $(1 - \alpha)$ probability of choosing the optimal action. As such, we can reduce the guide sampling policy by $(1 - \alpha)$, such that the new guide sampling rate is $\alpha - (1 - \alpha)$. We can continue to apply the procedure until the guide sampling rate $\alpha = 0$, as per Algorithm 1. The assumption of convergence between steps of $\alpha$ is key to the success of GRL. In complex environments, it can be very difficult to guarantee this convergence, as shown in Figure 4. This challenge motivates the introduction of a roll-back mechanism, in our proposed GRL-RB described below.

#### 3.1.2 METHOD: IMPLEMENTATION AND SETTINGS

We implemented our proposed GRL (Algorithm 1) on top of the Implicit Q-Learning algorithm (Kostrikov et al., 2022). We used the Clean Offline Reinforcement Learning (CORL) (Tarasov et al., 2023) implementation of IQL[1]. IQL was chosen for all of our experiments, to ease the transition from offline-to-online training for the AntMaze experiments in Section 4. The Combination Lock environment does not have offline data, so instead the guide was implemented as an oracle, able to determine the next number in the sequence, given the current observation. As such, IQL was only

---

[1] https://github.com/tinkoff-ai/CORL/blob/main/algorithms/finetune/iql.py

used in online training mode for these experiments. For variation 3, which uses a negative dense reward, we used the challenging AntMaze tasks from the D4RL benchmarks (Fu et al., 2020), which involve an 8-DoF quadraped robot attempting to reach a goal square in a maze environment. We ran experiments for AntMaze-Umaze-v2 with a reward of $r = -1$ for each time step until the Ant reaches the goal. The episodes automatically terminate at 700 time steps if the Ant fails to find the goal. Hyperparameter specifications can be found in Appendix F.

**Comparison Algorithms**   We designed three comparison algorithms based on the methods discussed in Section 2. Two *static* guide algorithms are generated (based on Chang et al. (2015), Chang et al. (2023), Daumé et al. (2009)), which sample the guide for 25% and 75% of the time throughout the entirety of training ('S25%' and 'S75%' in plots respectively). Following one of the approaches from (Yang et al., 2022), a *dynamic* linear decay method ('LD' in plots) is also introduced, which decreases the use of the guide agent from 100% to 0% by $\frac{1}{n}$% every evaluation, where $n$ is the number of evaluations during training. Our experiments found that this problem was too challenging for standard RL. Despite the conceptual simplicity of the Combination Lock environment, the agent is required to correctly choose a digit between 0-9 ten times in a row (a $10^{-10}$ probability of doing it by chance). We tested an implementation of PPO (Raffin et al., 2021), however it (unsurprisingly) made no progress in $1e^5$ time steps. Similarly, we used an implementation of vanilla SAC Raffin et al. (2021) for the AntMaze environments, and despite trying 100 different hyperparameter combinations per environment (using hyperparameter tuning framework Optuna (Akiba et al., 2019)), we found it could not improve without the guidance from the offline training component.

### 3.2   Results and Experimental Validation

We present results for the sampling rate for three variations and their experimental validation.   For completeness, Appendix E presents the full derivations.

#### 3.2.1   Variation 1: Optimal Guide Policy

For an optimal guide policy, $a_g = a^*$ every time step that $a \sim \pi_g$. We conservatively assume that every learner action $a_l$ is non-optimal $a_l = \overline{a^*}$, thus instantly ending the episode. While $\pi_l$ (which would usually be randomly initialised) is unlikely to be this poor, this assumption represents the worst case scenario. Thus, the sampling rate $\alpha$ represents an upper bound on the rate required to guarantee Equation 2. Every trajectory $\tau$ is a sequence of optimal guide actions (unless the first action is $a \sim \pi_l$) taken with probability $\alpha$, followed either by a learner action $a_l$ that then ends the episode prematurely (at $t = h$), or a guide step that completes the episode successfully (at $t = H$). While the result is a $>=$ relation (as per Equation 2), choosing an $\alpha$ that is larger will simply result in a slower transfer to the $\pi_l$, so we substitute it for $=$ here.

**Result.**   The derived sampling rate for an optimal $\pi_g$ and a terminating $\pi_l$ is:

$$\alpha = \mu^{\frac{1}{H}} \tag{3}$$

Equation 3 is used in the GRL algorithm, which is then applied to the Combination Lock environment, with a return degradation threshold $\mu = 0.75$ (Figure 1). As expected, the mean score of the agent was successfully maintained above the threshold $\mu \hat{R}_{\pi_g}$, and quickly reached the original guide score. In this case, $\pi_g$ was already optimal, however this procedure allows the learning agent to explore for a better solution without resulting in excessive performance degradation. $(1 - \alpha)$ increases smoothly throughout training until control is transitioned fully to $\pi_g$. While the LD is somewhat similar to GRL, it does not use the performance degradation-based $\alpha$, and so runs the risk of being overly-conservative,

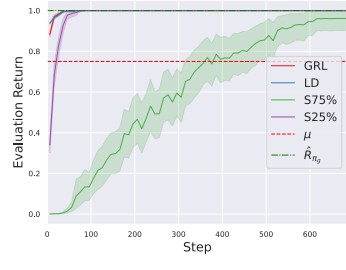

(a) Return

(b) Mean evaluation sampling rate

Figure 1: Optimal guide policy results: (a) **Return**.   The chosen return degradation threshold $\mu = 0.75 * \hat{R}_{\pi_g}$ is respected. $\hat{R}_{\pi_g}$ indicates the original score of the guide. Shading shows $1 - \sigma$ across 50 random seeds. (b) **Mean evaluation sampling rate** of the learning policy $\pi_l$ throughout training.

resulting in a slow transfer from $\pi_g$ to $\pi_l$. LD also decreases $\alpha$ every evaluation no matter the $\hat{R}_{\pi_g}$ result, meaning it could also transfer too quickly. In this case, however, with an optimal $\pi_g$, LD is able to perform well. The static guidance clearly breaches the performance degradation threshold, as the sampling rate is far lower than $\alpha$. In Figure 1b, S25% and S75% sometimes do not adhere strictly to their respective 25% and 75% sampling rates for two reasons. Firstly, given there are 10 steps in this particular problem, the exact sampling rates cannot be attained. Secondly, since the episode ends following a non-optimal action, $1 - \alpha$ can exceed the given rate if the first action is chosen to be a learner action.

### 3.2.2 VARIATION 2: NON-OPTIMAL GUIDE POLICY AND NON-TERMINATING LEARNING POLICY

We will now assume that the guide policy $\pi_g$ only takes the optimal action $a^*$ with probability $\beta_g < 1$. We will also assume that the learning policy does not immediately terminate the episode, but only takes the non-optimal action $\overline{a^*}$ with probability $\beta_l < 1$.

**Result.** The derived sampling rate for a non-optimal $\pi_g$ and a non-terminating $\pi_l$ is:

$$\alpha = \frac{\mu^{\frac{1}{H}}\beta_g - (1 - \beta_l)}{\beta_g - (1 - \beta_l)} \quad (4)$$

It is easy to see that if the learning policy is always terminating, i.e. $\beta_l = 1$, then we immediately recover Equation 3. Moreover, if a practitioner does not know $\beta_g$ and knows only the empirical mean evaluation return $\hat{R}_{\pi_g}$, they can easily use the relationship in Equation 16 to substitute in for $\beta_g$. Writing $R_{\pi*} = r$:

$$\beta_g = \left(\frac{\hat{R}_{\pi_g}}{R_{\pi*}}\right)^{1/H} \quad (5)$$

Depending on the environment, $\beta_l$ might be difficult to determine or the criticality of the application might force the practitioner to assume $\beta_l = 1$, representing the worst case scenario. If the application has a little more flexibility, then a lower $\beta_l$ would reduce the time needed to transfer from $\pi_g$ to $\pi_l$.

The experimental validation of the result for a non-optimal guide and non-terminating learning policy was run by forcing $\pi_g$ to take a random action with $1 - \beta_g$ probability, and fixing the $\pi_l$ action to the correct choice with $1 - \beta_l$ probability. The parameters for these experiments are given in Appendix F.2 and the results are shown in Figure 2. Again, the degradation threshold is respected by GRL, and online exploration of the space allows the learning agent to quickly surpass the mean evaluation score of the non-optimal guide agent, which starts at a low level of performance ($\hat{R}_{\pi_g} \approx 0.35$). While the degradation threshold is also respected by LD, the non-optimal guide demonstrates the tendency for LD to be over-conservative, whereas GRL makes use of the derived $\alpha$ to transfer as fast as possible while respecting the degradation threshold.

### 3.2.3 VARIATION 3: DENSE REWARD

We now consider two cases of a dense reward. The first is a positive step-wise reward $r_t = r$ ($r > 0$) for each non-terminal action taken at time step $t$. Environments that may use such a reward are those where an agent must choose actions to continue a task as long as possible, such as the aforementioned Combination Lock, or the classic control problem, CartPole. The latter is a negative step-wise reward $r_t = r$ ($r < 0$) for each non-terminal action taken at time step $t$, which could be applicable in environments where an agent must complete as task as quickly as possible, such as navigating a maze.

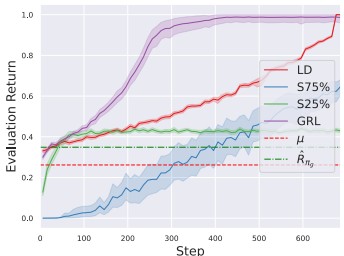

(a) Return

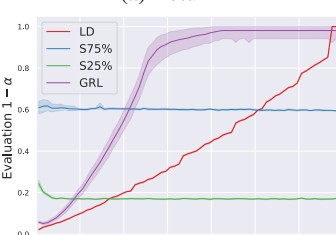

(b) Mean evaluation sampling rate

Figure 2: Non-optimal guide and non-terminating learner results: (a) **Return**. The chosen return degradation threshold $\mu = 0.75 * \hat{R}_{\pi_g}$ is respected. The $\hat{R}_{\pi_g}$ line indicates the original score of the guide. (b) **Mean evaluation sampling rate** of the learning policy $\pi_l$ throughout training.

**Result: Negative Dense Reward.** The derived sampling rate for a non-optimal $\pi_g$ and a non-terminating $\pi_l$ with a negative step-wise (dense) reward is:

$$\alpha = \frac{\mu^{\frac{1}{H}} - (1 - \beta_l)}{\beta_g - (1 - \beta_l)} \quad (6)$$

**Result: Positive Dense Reward.** The derived sampling rate for a non-optimal $\pi_g$ and a non-terminating $\pi_l$ with a positive step-wise (dense) reward is:

$$\sum_{h=1}^{H} (\beta')^h (1 - \beta')^{1-\delta(H-h)} hr_t - \mu \hat{R}_{\pi_g} \geq 0 \quad (7)$$

where $\beta' = \alpha\beta_g + (1 - \alpha)(1 - \beta_l)$.

Given that the AntMaze environment definitely aligns to the non-terminating learner variation, it can be difficult to choose an appropriate $\beta_l$ as per Section 3.2.2. We chose $\beta_l = 0.1$ to be conservative, however given the far longer time between evaluations in AntMaze compared to Combination Lock, and the excellent warm-start form the guide, the agent is able to improve significantly even after just one update. Because of this, the online learner in Figure 3a degrades to nowhere near the threshold, we could have chosen an even riskier $\beta_l$. The parameters for these experiments are shown in Appendix F.3 and the results in Figure 3. Again, we see an overly-conservative LD and S25%, while S75% shows how overuse of $\pi_l$ during initial training can affect performance.

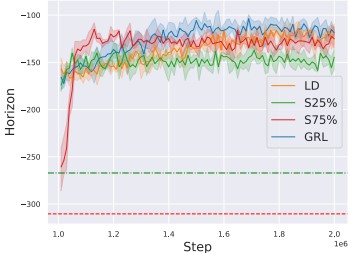

(a) Return

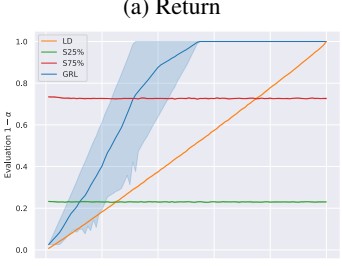

(b) Mean evaluation sampling rate

Figure 3: Negative Dense Reward Results: (a) **Return.** The chosen return degradation threshold with $\mu = 0.9$ is respected. The $\hat{R}_{\pi_g}$ line indicates the original score of the guide. (b) **Mean evaluation sampling rate** of the learning policy $\pi_l$ throughout training.

## 4 GUIDED REINFORCEMENT LEARNING WITH ROLL-BACK (GRL-RB)

As discussed above, the assumption of convergence between updates of $\alpha$ ensures the success of GRL. However, if $\pi_l$ has not yet converged such that we can be confident it is sampling the optimal action with $1 - \alpha$ probability, then decreasing $\alpha$ for the next step of training may cause GRL to over-sample the non-optimal action and cause the return to fall below Equation 2. To help the GRL algorithm recover from non-convergence, and minimise the number of evaluations that fall below the defined threshold, we introduce the Guided RL with Roll-Back (GRL-RB) (Algorithm 2, described fully in Appendix G). In GRL-RB, if the evaluation return falls below Equation 2, then $\alpha$ is 'rolled back' to the value that attained the previous best return. Beyond maintaining good online performance, another benefit of the GRL-RB approach is to speed up the transfer of learning from $\pi_g$ to $\pi_l$. GRL-RB allows the algorithm to take larger steps of $\alpha$ (and more frequently), while minimising degradation through the roll-back mechanism.

As a straightforward demonstration of the value of GRL-RB, we make intentionally poor hyperparameter choices for GRL in the Combination Lock environment. This makes convergence unlikely, and we can immediately see how GRL struggles to respect the chosen threshold (Figure 4). By comparison, GRL-RB gives the learner the opportunity to return to the previous best-performing $\alpha$, and recover. Additional experimental results in this environment are provided in Appendix F. Experiments with the algorithms in a more challenging environment, Antmaze, are shown in Figure 5. We compare GRL-RB with JSRL (Uchendu et al., 2023) and another offline-to-online method, IQL (Kostrikov et al., 2022). See Appendix G.1.2 for details. To demonstrate GRL-RB's flexibility to transfer hyperparameters, we include a robustness experiment in Figure 9 in Appendix G.1.3.

These results shown the roll-back mechanisms effectiveness in helping GRL to maintain the performance threshold with non-ideal hyperparameter choices, and in challenging environments. In Figure 4, we show the maximum and minimum score attained across 50 runs to show all threshold violations. While LD is more effective in the beginning with small $1 - \alpha$ values, it can also be pushed

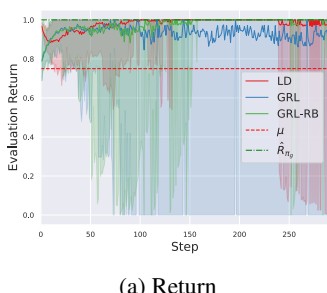
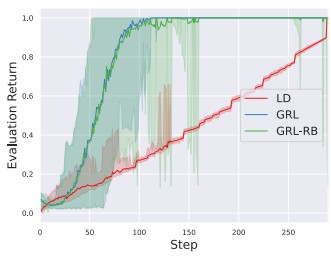

(a) Return                      (b) Mean evaluation sampling rate

Figure 4: GRL, GRL-RB and LD in the Combination Lock environment, under variation 1, with an intentionally poor choice of hyperparameters. Shading shows the maximum and minimum score attained across all runs to highlight the performance threshold violations. (a) **Return**. The online training return. (b) **Mean evaluation sampling rate** of the learning policy $\pi_l$ throughout training.

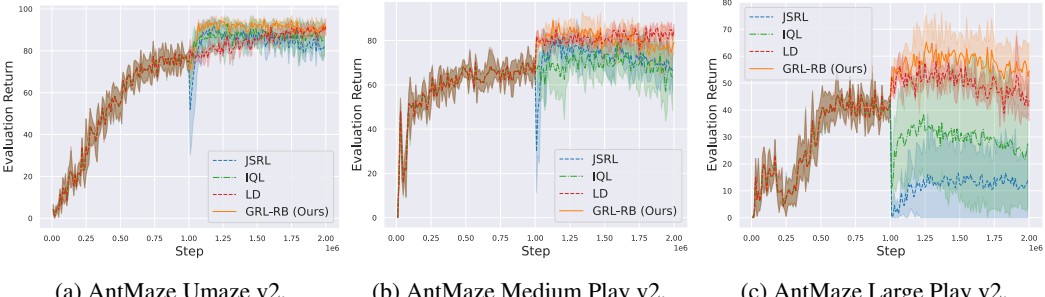

(a) AntMaze Umaze v2.       (b) AntMaze Medium Play v2.       (c) AntMaze Large Play v2.

Figure 5: Results for the Antmaze environments. Offline pre-training phase at step $< 1e6$. The same pre-trained policy is then used for all algorithms as the online guide agent (step $> 1e6$). While the stepwise penalty $r = -1$ (from Section 3.2.3) was used as the guide's evaluation score and to calculate the performance degradation threshold, we report the standard normalized AntMaze scores for better comparison with the literature.

onto the next stage before convergence (as $1 - \alpha$ is increased no matter the evaluation score), so its performance can deteriorate at higher $1 - \alpha$ values. GRL achieves a reasonably good average results, but clearly starts being challenged around 50 time steps, and cannot recover effectively. Around time step 125 by comparison, we see GRB-RL start to struggle, however its $\alpha$ is immediately rolled back and its return quickly climbs above the threshold. We see further evidence of GRL-RB's ability to effectively handle the trade-off between transfer speed and performance in Figures 8e and 8f, where even though GRL achieves a faster transfer, the return achieved by GRL-RB is better throughout training, as it self-adjusts to adhere to the threshold as required. Figures 5 and 9 show that guided policy methods that do not consider performance degradation struggle in the initial stages of transfer (even if they recover eventually in simpler environments, e.g. Figure 5a) so mechanisms like the roll-back are needed for applications where it is critical to maintain good performance.

## 5 DISCUSSION

Transferring from $\pi_g$ to $\pi_l$ is a careful balance between speed and managing performance degradation. For complex tasks, $\pi_l$ may take time to learn the optimal behaviours in the environment, improving upon the behaviour of $\pi_g$. However, an extremely slow transfer may negate some of the benefits of warm-starting learning using a guide, and so ideally we would like $\pi_g$ to transfer as fast as possible to $\pi_l$, while assuring some level of performance. In our first contribution, GRL, we introduced an algorithm that enabled a successful transfer from $\pi_g$ to $\pi_l$. This algorithm is simple to implement on top of existing algorithms (e.g., IQL) and does not require any additional training compared to vanilla RL, as $\pi_g$ stays static throughout learning. As $\pi_g$ is sampled any time throughout an episode, it avoids using the learner for many consecutive time steps, which can take the agent so far from an optimal trajectory that it cannot recover (e.g. Figure 6). The user is free to set the initial sampling rate $\alpha$ of GRL, and so in our second contribution, we show how this can be chosen most advantageously.

$\pi_l$ does need to explore to some extent, which will inevitably result in some level of degradation, however for many applications it is possible to specify an acceptable level of exploration degradation. In our second contribution, we have sought to take advantage of this specified performance degradation threshold. Our theoretical results in Section 3.2 show $\alpha$ can be chosen in certain environments to enable the fastest transfer possible while respecting performance requirements. LD is reasonably effective but can be over-conservative and slow to transfer. The static $\pi_g$ sampling algorithms (S25% and S75%) are clearly unable to converge unless $\pi_g$ is already optimal, so it is surprising that this method continues to be used in the guided policy literature (Section 2). Our analysis has shown that GRL is the only method that respects the performance degradation threshold in all experiments.

In an ideal situation, GRL alone would be sufficient to enable knowledge transfer from $\pi_g$ to $\pi_l$. The result in Figure 8a shows clearly that when the policy is able to fully converge between updates, the roll-back mechanism introduces no additional advantage to the algorithm. When convergence is not assured however, our third contribution, GRL-RB, is able to adaptively adjust its sample rate to better respect the performance degradation threshold. The results in Figure 8 (Appendix G.1.1) show the strong utility of the roll-back mechanism of GRL-RB in alleviating these issues. Specifically, Figure 8c-8h show that rolling back the sample rate allows the performance to recover. In Figure 8d, it is clear to see the learner sampling rate is dropped automatically at time steps above 250, due to the agent's drop in evaluation score at the corresponding time steps as in Figure 8c. The agent's score then gradually increases until it returns to its previous optimal value. On average, GRL-RB is able to maintain performance above the threshold even with obviously poor choices of hyperparameters.

The AntMaze results in Section 4 further demonstrate the utility of both GRL and GRL-RB, in a more complex environment. While IQL without any modification does very well when it has a variety of data available in its replay buffer (Table 6), it struggles when the replay buffer is cleared for online learning. This makes it unsuitable to applications without a dataset available (e.g., when $\pi_g$ is a set of rules). Furthermore, the same policy is shifted from offline to online learning, meaning it must be of the same format (i.e., a neural network). JSRL also gets some reasonable results, however it suffers from the horizon-based action-selection, as discussed in Section 2. It can struggle to recover from the initial overuse of the learner agent in the more challenging environments (medium and large). Conversely, performance degradation does not occur in these environments for GRL-RB or LD, as our percentage-based sampling method ensures the restricted use of the learner. Even in the extreme case presented for the additional robustness experiment in Appendix G.1.3, the GRL-RB method is effective in preventing degradation by constantly rolling back the sampling rate when required.

A limitation of GRL-RB is that the roll-back is only triggered once the score has fallen below the threshold. A valuable addition would be a mechanism to predict or detect whether this is about to occur, to prevent the violation altogether. Further, future work could make use of a statistical method to compare the evaluation score distributions (as in Dasagi et al. (2019)), to provide an enhanced measure of evaluation score improvement. Finally, the derivations of $\alpha$ provided above have a fairly limited scope compared to the wide variety of environments and reward schemes used in RL. In future work, we will aim to derive more results for other kinds of environments.

## 6 CONCLUSION

We introduce two algorithms, GRL and GRL-RB, to enable the transfer of prior knowledge from a guide policy $\pi_g$ to learning policy $\pi_l$. Our theoretical analysis and experimental confirmation show that for environments with certain reward schemes, an initial sample rate $\alpha$ for the $\pi_g$ can be derived to ensure the mean evaluation score does not fall below a user-defined threshold. This is important, as many systems that currently use classical controllers may require to maintain a certain evaluation score, due to reasons including safety, cost or user experience. It is thus critical that algorithms designed to replace such systems have some guarantee of performance. To the best of our knowledge, this is the first time a performance guarantee has been established for a guided RL method.

The sample rates we derive do have a requirement that the agent is able to fully converge to the original evaluation score between evaluations, before $\alpha$ is progressed. Choosing the correct hyperparameters to assure this can be challenging, so we introduced GRL-RB to allow the agent the opportunity to return to a previous best $\alpha$. Results in the Combination Lock and AntMaze environments show that GRL-RB enables effective self-correction, providing some flexibility for use in more complex environments.

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

## A ADDITIONAL RELATED WORK

There are many approaches in the literature to providing an agent with knowledge of a task before it commences online learning. While the most relevant approaches to our algorithm are discussed in Section 2, ere we summarise approaches that influence the replay buffer, parameter updates, and initial parameters of an agent.

### A.1 REPLAY BUFFER

The replay buffer approach is primarily found in algorithms using the learning from demonstrations (LfD) method (Rajeswaran et al., 2017; Vecerik et al., 2017; Hester et al., 2018; Wang et al., 2022a; Nair et al., 2018; Goecks et al., 2019). In LfD, trajectories generated from a teacher (e.g. a human expert, or a partially trained agent) are made available to the agent during training (through the replay buffer) to provide examples of high quality behaviour. This has found to be effective even with a minimalistic application, by using a distribution of (Kakade & Langford, 2002), a single (Salimans & Chen, 2018), or the most significant (Tavakoli et al., 2019) demonstration(s) as states from which to restart agent exploration. These significant states could be even be found through online agent exploration, then archived for learning through LfD methods (Ecoffet et al., 2021). For goal-oriented tasks, a curriculum of starting states can be produced, increasingly further from a single demonstrated state that achieves the goal (Florensa et al., 2017). The demonstrations can help to emphasise the direction in which the agent's parameters should be updated to solve its task, or to guide the agent to states covered by an optimal (or near-optimal) policy. A unique approach in (Wu et al.) learned an additional Planner policy, which could sample natural-language based instructions to give to the Actor as part of the input to its policy network. The rewards in this case for both the planner and actor were also modified to encourage the actor to follow the planner guidance.

### A.2 PARAMETER UPDATES

Modifying the parameter update typically involves adding an additional term to the loss or policy gradient. It is common for Replay Buffer-based methods to also modify the parameter update, to encourage imitation of the demonstrated trajectories. For example, the (Nair et al., 2018) adds a behaviour-cloning based term to expert actions that appear optimal, (Rajeswaran et al., 2017) introduces a term to weight contributions from advantageous demonstrations to the gradient, and (Hester et al., 2018) uses a margin function to increase the q-values of the expert actions. in Levine & Koltun (2013), Guided Policy Search instead uses a dynamic programming approach to generate multiple guide policies, whose samples are used to update the online learner. An additional importance weighting term is also introduced to better incorporate the off-policy samples to the policy gradient-based algorithm.

### A.3 INITIAL PARAMETERS

Methods that influence the initial parameters of the agent's policy and/or value function typically involving a portion of offline pre-training. This is advantageous when there is a batch of data available, and thus batch RL (Levine et al., 2020) (also known as offline RL) techniques can be used to warm-start learning. While this is an appealing approach - to directly take the behaviours the agent has learned from the restricted batch of data, then continue to improve it online - this approach faces challenges with performance degradation. This issue was analysed in Wexler et al. (2022) and found to be due to overestimation bias, similar to that experienced in offline RL (Fujimoto et al., 2019)(Fujimoto & Gu, 2021). Thus, such methods typically address this in their parameter update. For example, in Wexler et al. (2022), a penalty is introduced to the policy objective to reduce the size of gradient updates when their measure of extrapolation error is high. (Xu et al., 2022) uses a $min$ function over Q-values from multiple experts to reduce overestimation of Q-values. The aforementioned (Hester et al., 2018) also uses a pre-training method, and the margin function acts to ensure Q-values of unexplored actions will not be estimated as higher than the experts' actions. Some methods do not explicitly address performance degradation, but instead load a pre-trained critic along with the policy (Goecks et al., 2019)(Uchendu et al., 2023), which can help to avoid degradation (Wexler et al., 2022).

The initial parameters could also be influenced using an adaptive rules-based approach. The methods in Chen et al. (2023)(Wang et al., 2022b)(Shi et al., 2022)(Yu et al., 2023) encode the knowledge as rules with adaptive parameters, which can be refined via online RL. (Silva & Gombolay, 2021) instead encodes rules as a differential decision tree, whose parameters are also adaptable through online training. These methods can be useful when a policy can be easily encoded as a rule set with obvious choices for weighting parameters or membership functions, however can be challenging for more complex decision policies.

## B  EXAMPLE OF JSRL FAILURE MODE

The JSRL horizon-based curriculum, where the learner $\pi_l$ takes over the action-selection from the guide policy $\pi_g$ after time step $h$, may have issues in any environment with variable length episodes. An example is shown in Figure 6, where $\pi_g$ (red) controls actions for most of the path to Goal. However, the episode runs over time due to the non-optimal path around coordinate $(7, 5)$. At this point $t > h$, so $\pi_l$ takes over (blue), but it is too far from the goal to succeed.

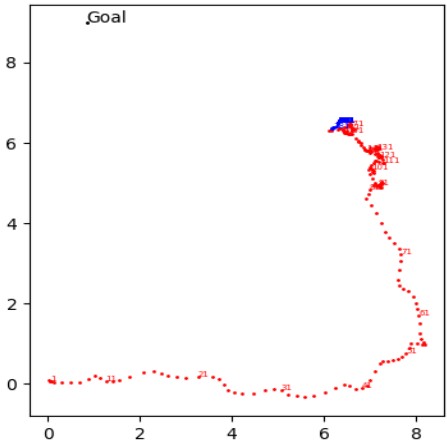

Figure 6: A failed episode of JSRL in the AntMaze-Umaze environment.

## C  COMBINATION LOCK ENVIRONMENT

The combination lock environment CombinationLock-v0[2] represents a simple MDP where an agent must choose the correct next number in the combination. It is represented in Figure 7.

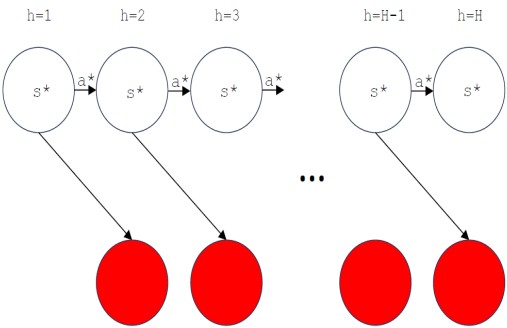

Figure 7: A simple Combination Lock MDP.

---

[2]GitHub link to be included in final version.

It receives a sparse reward of 1 if it reaches the end of the combination, otherwise an incorrect choice will immediately end the episode.

If the agent reaches the end of the combination, it receives a sparse reward of $r$:

$$r_t = \begin{cases} r & \text{if } t = H - 1 \text{ and } a_t = a* \\ 0 & \text{otherwise} \end{cases} \tag{8}$$

The environment takes a parameter $H$ allowing the user to specify the desire length of the combination. In our experiments, $H = 10$. The combination does not change between episodes (this would be impossible to learn), and so the combination for our experiments was set to $[0, 1, 2, 3, 4, 5, 6, 7, 8, 9]$.

The initial observation space is a vector of -1 of length $H$. The action space can be discrete or a continuous vector of length 10 (representing digits 0-9). The environment will apply an $\arg\max$ to the continuous action vector to determine the digit - this was the setting used for our experiments. If the agent-supplied action is the incorrect digit, the episode will terminate with a reward $r = 0$. If the action is correct, and $t \neq horizon$ the observation vector will update with the correct digit replacing the -1, and giving a reward $r = 0$. If $t = H$, the agent receives $r = 1$ and the episode terminates.

## D GUIDED REINFORCEMENT LEARNING ALGORITHM

The GRL algorithm is shown in Algorithm 1.

---

**Algorithm 1** Guided RL Algorithm

---

**Require:** guide sample rate $\alpha$, guide policy $\pi^g$, evaluation frequency $k$

1: Initialise learning policy $\pi^l$ and Q-function $\hat{Q}$ from scratch, replay buffer $B \leftarrow \emptyset$, $\hat{R}_{\pi_g} \leftarrow$ mean evaluation return of $\pi^g$, $n_{\pi_l} \leftarrow 0$, current guide sample rate $\alpha_c \leftarrow \alpha$
2: **while** $\alpha_c > 0$ **do**
3:   **for all** episode **do**
4:     **for all** episode step $t$ **do**
5:       observe state $s_t$
6:       $\beta \leftarrow$ uniform random number between 0 and 1
7:       **if** $\beta < \alpha_c$ and $\frac{n_{\pi_l}}{t} < \alpha_c$ **then**
8:         $a_t \leftarrow \pi^g(s_t)$
9:       **else**
10:         $a_t \leftarrow \pi^l(s_t)$
11:         $n_{\pi_l} \leftarrow n_{\pi_l} + 1$
12:       observe $r_t, s_{t+1}$
13:       $B \leftarrow B \cup (s_t, a_t, r_t, s_{t+1})$
14:       $\pi^l, \hat{Q} \leftarrow Train(\pi^l, \hat{Q}, B)$
15:       **if** $t\%k == 0$ **then**
16:         $\hat{R}_\pi \leftarrow$ mean evaluation return for $\pi$
17:         **if** $\hat{R}_\pi >= \hat{R}_{\pi_g}$ **then**
18:           $\alpha_c \leftarrow \alpha_c - (1 - \alpha)$

---

## E SAMPLING RATE DERIVATIONS

We will refer to an action chosen from $\pi_g$ as $a_g$, and from $\pi_l$ as $a_l$. The Combination Lock MDP (Appendix C) is summarised in the following list:

1. Deterministic dynamics: $p(s_{t+1}|s_t, a_t) = 1$
2. Deterministic start state: $\rho_0(s_0) = 1$
3. State space: $s \in [s^*, \overline{s^*}]$, where $s^*$ is a state visited by the optimal policy and $\overline{s^*}$ is terminal
4. Action space: $a \in [a^*, \overline{a^*}]$, where $a^*$ is an optimal action
5. Rewards: Sparse reward of 1 for successful episode. See Equation 8.

6. Policy: $\pi(a_g|\cdot) = \alpha$, $\pi(a_l|\cdot) = 1 - \alpha$

Incorporating the above assumptions, and letting $R_\tau = \sum_0^{T-1} \gamma^t r_t$ for simplicity. Equation 1 can be simplified to:

$$\hat{R}_\pi = \sum_\tau \left( \prod_{t=0}^{H-1} \pi(a_t|s_t) R_\tau \right) \tag{9}$$

Incidentally, if the minimum possible score in the environment is 0, then Equation 2 simplifies to:

$$\hat{R}_\pi >= \mu \hat{R}_{\pi_g} \tag{10}$$

## E.1 VARIATION 1: OPTIMAL GUIDE AND TERMINATING LEARNER.

Assume $a_g = a^*$ for each $t$ where $a \sim \pi_g$, and every learner action $a_l$ is non-optimal $a_l = \overline{a^*}$, thus instantly ending the episode. Equation 9 can thus be written as:

$$\hat{R}_\pi = \sum_{\tau_h} \alpha^{h-1} \alpha^{\delta(H-h)} (1-\alpha)^{1-\delta(H-h)} R_{\tau_h}$$

$$= \sum_{h=0}^H \alpha^h (1-\alpha)^{1-\delta(H-h)} R_{\tau_h} \tag{11}$$

where $\delta$ is the unit impulse function. For the sparse reward case, given $r_t = 0$ for every time step $t < H$, and thus $R_\tau = 0$ for every trajectory $\tau$ other than the one that reaches the sparse reward at $t = H$, that trajectory is the only term left in the sum. Equation 11 then simplifies to:

$$\hat{R}_\pi = \alpha^H (1-\alpha)^{1-1} r$$

$$= \alpha^H r \tag{12}$$

We can now solve for $\alpha$, by substituting this result (Equation 12) into Equation 10.

$$\hat{R}_\pi >= \mu \hat{R}_{\pi_g}$$

$$\alpha^H r >= \mu r$$

$$\alpha >= \mu^{\frac{1}{H}} \tag{13}$$

## E.2 VARIATION 2: NON-OPTIMAL GUIDE AND NON-TERMINATING LEARNER

Assume $\pi_g$ takes $a^*$ with probability $\beta_g < 1$, and that $\pi_l$ does not immediately terminate the episode, but takes $\overline{a^*}$ with probability $\beta_l < 1$. This means the probability of this policy sampling the optimal or the non-optimal policy is:

$$[\pi(a^*|s), \pi(\overline{a^*}|s)] = [\alpha\beta_g + (1-\alpha)(1-\beta_l), \alpha(1-\beta_g) + \beta_l(1-\alpha))] \tag{14}$$

This is a simple modification to the previous variation, as still the only trajectory that generates a non-zero score is the one that reaches the end of the combination. As such, the new score is:

$$\hat{R}_{\pi_g} = (\alpha\beta_g + (1-\alpha)(1-\beta_l))^H r \tag{15}$$

Now, for a sparse reward the $\pi_g$ score will be the fraction of the optimal return corresponding directly to $\beta_g^H$:

$$\hat{R}_{\pi_g} = \beta_g^H r \tag{16}$$

As such, if we only consider a non-optimal guide policy, the $\beta_g^H$ will cancel and the result will be identical to Equation 13. This is not the case for $\beta_l$ which only influences the online return, and thus does not cancel with terms in $\hat{R}_{\pi_g}$. In this case, the corresponding $\alpha$ would be:

$$(\alpha\beta_g + (1-\alpha)(1-\beta_l))^H r >= \mu \hat{R}_{\pi_g} \tag{17}$$

$$(\alpha\beta_g + (1-\alpha)(1-\beta_l))^H r >= \mu \beta_g^H r$$

$$\alpha >= \frac{\mu^{\frac{1}{H}} \beta_g - (1-\beta_l)}{\beta_g - (1-\beta_l)}$$

### E.3 VARIATION 3: DENSE REWARD

This is a slightly more challenging setting, given that we not cannot discount trajectories that do not reach $t = H$. If we define the probability of taking the optimal action (from Equation 14) as:

$$\beta' = \alpha\beta_g + (1 - \alpha)(1 - \beta_l) \tag{18}$$

#### E.3.1 POSITIVE DENSE REWARD

Then Equation 9 becomes:

$$\hat{R}_\pi = \sum_{h=1}^{H-1} (\beta')^h (1 - \beta')^{1-\delta(H-h)} hr \tag{19}$$

We can still ignore the situation when $h = 0$, since $R_\tau = 0$ in this case. To get an appropriate sampling rate for the guide policy we have:

$$\sum_{h=1}^{H} (\beta')^h (1 - \beta')^{1-\delta(H-h)} hr_t - \mu\hat{R}_{\pi_g} \geq 0 \tag{20}$$

This expands out into a polynomial. Here is an example for $H = 4$:

$$\beta'(1 - \beta')r + (\beta')^2(1 - \beta')(2r) + (\beta')^3(1 - \beta')(3r) + (\beta')^4(4r) - \mu\hat{R}_{\pi_g} \geq 0$$
$$r((\beta')^4 + (\beta')^3 + (\beta')^2 + (\beta')) - \mu\hat{R}_{\pi_g} \geq 0 \tag{21}$$

Since $r > 0$ (as defined above), and thus $\hat{R}_{\pi_g} > 0$ (since $\hat{R}_{\pi_g}$ is a sum of $r$) and clearly $\mu > 0$, there will only be one sign change in this polynomial, and thus one positive root (representing the desired $(\beta')$) as per Descartes' law of signs. While there is no general expression for the solution (above $H > 4$), it is simple to use one of many numerical approximation methods such as the Newtown-Raphson method to find the positive root, and thus $\alpha$.

#### E.3.2 NEGATIVE DENSE REWARD

Many environments instead use a dense negative reward (penalty) $r < 0$ every time step to encourage agents to finish a task quickly. In this case, the total episode reward is negatively correlated with the time step. This case is actually as simple as the sparse reward case for episodic tasks. Such tasks will generally specify a maximum number of steps $N_{max}$ to complete the task. If we assume the guide agent completes the task in $N$ time steps (with probability $\beta_g$), while a learning agent's action causes the episode to run for $N_{max}$ time steps (with probability $\beta_l$), then similar to the sparse reward case, there are only two options for the total episode reward: $-rN$ or $-rN_{max}$. Using the probability for the optimal action $\beta'$ as above, and writing the probability for the non-optimal action as $(1 - \beta')$:

$$\hat{R}_\pi = (-rN)\beta'^H + (-rN_{max})\left(1 - \beta'^H\right) \tag{22}$$

We require the full Equation 2 for this setting, since the episode reward is now bounded between $[-rN_{max}, \hat{R}_{\pi_g}]$, so:

$$\hat{R}_\pi >= -rN_{max} + \mu\left(rN_{max} - rN\right) \tag{23}$$

Substituting the left hand side of Equation 22 into Equation 23 gives:

$$(-rN)\beta'^H + (-rN_{max})\left(1 - \beta'^H\right) >= -rN_{max} + \mu\left(rN_{max} - rN\right)$$
$$\beta'^H(rN_{max} - rN) - rN_{max} >= -rN_{max} + \mu\left(rN_{max} - rN\right)$$
$$\beta' >= \mu^{\frac{1}{H}} \tag{24}$$

as for the sparse reward case. Substituting back in for $\beta'$:

$$\alpha = \frac{\mu^{\frac{1}{H}} - (1 - \beta_l)}{\beta_g - (1 - \beta_l)} \tag{25}$$

# F  Guided RL Algorithm (GRL) - Experiments

For all of the experiments with the GRL algorithm, we use the default values for IQL's hyperparameters, except for those specified in the Tables 1 and 2. All experiments in this paper were run on a machine with 2 Nvidia GeForce RTX 4090 GPUs, and 64 CPUs with 512 GB of RAM. Each seed used approximately 5 GB of CPU RAM and 3 GB of GPU memory.

## F.1  GRL Variation 1 - Experiments

For variation 1, we evaluated a learner with an optimal guide policy in the Combination Lock environment across 10 random seeds, with the parameters shown in Table 1.

Table 1: Parameter variations from IQL defaults in the Combination Lock experiment with optimal $\pi_g$ and terminating $\pi_l$. An update frequency of 1 indicates one update to the policy per time step in the environment.

| $H$ | $\mu$ | $\beta_g$ | $\beta_l$ | Batch Size | Buffer Size | Evaluation Frequency | Update Frequency |
|---|---|---|---|---|---|---|---|
| 10 | 0.75 | 1.0 | 1.0 | 64 | 256 | 10 | 1 |

## F.2  GRL Variation 2 - Experiments

For variation 2, we evaluated a learner with an non-optimal guide policy in the Combination Lock environment across 10 random seeds, with the parameters shown in Table 2.

Table 2: Parameter variations from IQL defaults in the Combination Lock experiment with non-optimal $\pi_g$ and non-terminating $\pi_l$.

| $H$ | $\mu$ | $\beta_g$ | $\beta_l$ | Batch Size | Buffer Size | Evaluation Frequency | Update Frequency |
|---|---|---|---|---|---|---|---|
| 10 | 0.75 | 0.9 | 0.7 | 64 | 256 | 10 | 1 |

## F.3  GRL Variation 3 - Experiments

As discussed in Section 3.2.3, for this variation we used the AntMaze-Umaze-v2 environment (Fu et al., 2020). In this environment, the assumption that $\rho_0(s_0) = 1.0$ is violated, as the start position of the Ant is slightly randomised. However, we chose to use the 'play' rather than the 'diverse' environments to meet our assumptions as closely as possible, as the goal positions are also randomised in the diverse environments. A stricter degradation threshold $\mu = 0.9$ was used to calculate $\alpha$, since $\pi_g$ pre-trained offline using IQL reached a high $R_{\pi_g}$. Further, given the strong unlikelihood of a single learner step causing the ant to go completely off-track and attain the minimum reward, we set $\beta_l = 0.1$. We used IQL default hyperparameters with the exception of the variations listed in Table 3.

Table 3: Parameter variations from IQL defaults in the AntMaze experiments.

| $\beta$ | IQL $\tau$ | IQL Deterministic | Buffer Size | Evaluation Frequency | Evaluation Episodes | Online Iterations |
|---|---|---|---|---|---|---|
| 10 | 0.9 | False | 10000 | 10000 | 100 | 1000000 |

# G  Guided RL with Roll-Back Algorithm

The second contribution of this work is the GRL-RB algorithm. GRL-RB builds on GRL by introducing a 'roll-back' mechanism that allows the agent to decrease the sampling rate of the guide policy in the case of degradation of the evaluation score below a user defined threshold. The full algorithm is described in Algorithm 2.

---

**Algorithm 2** Guided RL with Roll-Back Algorithm

---

**Require:** degradation threshold $\mu$, guide sample rate $\alpha$, guide policy $\pi^g$, evaluation frequency $k$
1: Initialise learning policy $\pi^l$ and Q-function $\hat{Q}$ from scratch, replay buffer $B \leftarrow \emptyset$, $\hat{R}_{\pi_g} \leftarrow$
    mean evaluation return of $\pi^g$, current guide sample rate $\alpha_c \leftarrow \alpha$, $\alpha^* \leftarrow \alpha_c$, $R^* \leftarrow \hat{R}_{\pi_g}$,
    $rolled\_back \leftarrow False$ ;
2: **while** $\alpha_c > 0$ **do**
3:    **for all** episode **do**
4:       **for all** episode step $t$ **do**
5:          observe state $s_t$
6:          $n \leftarrow$ uniform random number between 0 and 1
7:          **if** $n < \alpha_c$ **then**
8:            $a_t \leftarrow \pi^g(s_t)$
9:          **else**
10:           $a_t \leftarrow \pi^l(s_t)$
11:         observe $r_t, s_{t+1}$
12:         $B \leftarrow B \cup (s_t, a_t, r_t, s_{t+1})$
13:         $\pi^l, \hat{Q} \leftarrow Train(\pi^l, \hat{Q}, B)$
14:         **if** $t\%k == 0$ **then**
15:           $\hat{R}_\pi \leftarrow$ mean evaluation return for $\pi$
16:           **if** $\hat{R}_\pi >= \hat{R}_{\pi_g}$ **then**
17:             **if** $rolled\_back$ **then**
18:               $\alpha_c \leftarrow \alpha_{rb}$
19:               $rolled\_back \leftarrow$ False
20:             **else**
21:               $\alpha_c \leftarrow \alpha_c - (1 - \alpha)$
22:             **if** $\hat{R}_\pi >= \hat{R}^*$ **then**
23:               $\alpha^* \leftarrow \alpha_c$
24:               $R^* \leftarrow \hat{R}_\pi$
25:           **else if** $\hat{R}_\pi <$Eq (2) **then**
26:             $rolled\_back \leftarrow$ True
27:             $\alpha_{rb} \leftarrow \alpha_c$
28:             $\alpha_c \leftarrow \alpha^*$

---

### G.1 GRL-RB Results

#### G.1.1 Combination Lock

In this section we evalute GRL-RB within the Combination Lock environment. The experiments in this section were designed to demonstrate the need for the roll-back mechanism in the non-ideal but more realistic case of non-convergence between steps of $\alpha$. The chosen parameters for these experiments are shown in Table 4. Because this is a fairly simple environment, to show non-convergence between evaluations we chose parameters which will clearly hinder RL training, such as a batch size of just 10 samples, with just one update between evaluations. Since we are concerned about preventing degradation below $\mu * \hat{R}_{\pi_g}$, we evaluate them not just on the mean episode reward, but also on the amount of evaluation steps spent below $\mu * \hat{R}_{\pi_g}$. These results are also shown in Table 5.

Plots showing the evaluation scores and $(1 - \alpha)$ are shown in Figure 8. Figure 8a, 8c, 8e and 8g show evaluation scores for different variations of guide and learning policies. The parameters for Figure 8a were chosen to *allow* full convergence between steps of $\alpha$, while the parameters for Figure 8c, 8e and 8g were chosen to *prevent* full convergence between evaluations. Figure 8b, 8d, 8f and 8h show the sampling rate of the learning agent throughout training for the subfigure on the same row.

Table 4: Parameter sets used for experiments comparing GRL-RB and GRL.

| Parameter Set | $H$ | $\mu$ | $\beta_g$ | $\beta_l$ | Batch Size | Buffer Size | Evaluation Frequency | Update Frequency |
|---|---|---|---|---|---|---|---|---|
| A | 10 | 0.75 | 1 | 1 | 64 | 256 | 10 | 1 |
| B | 10 | 0.75 | 1 | 1 | 10 | 64 | 1 | 1 |
| C | 10 | 0.75 | 0.9 | 1 | 10 | 64 | 1 | 1 |
| D | 10 | 0.75 | 0.9 | 0.5 | 10 | 64 | 1 | 1 |

Table 5: Comparison of GRL-RB and GRL over different parameter sets from Table 1. 1-$\sigma$ uncertainties are shown.

| Parameter Set | GRL | | GRL-RB | |
|---|---|---|---|---|
| | Final Score | Evaluations below $\mu * \hat{R}_{\pi_g}$ | Final Score | Evaluations below $\mu * \hat{R}_{\pi_g}$ |
| A | $1.0 \pm 0.0$ | $0.0 \pm 0.0$ | $1.0 \pm 0.0$ | $0.0 \pm 0.0$ |
| B | $0.92 \pm 0.27$ | $4.0 \pm 11$ | $\mathbf{1.0 \pm 0.0}$ | $1.2 \pm 2.1$ |
| C | $0.84 \pm 0.37$ | $21 \pm 20$ | $\mathbf{1.0 \pm 0.0}$ | $\mathbf{2.9 \pm 3.5}$ |
| D | $1.0 \pm 0.0$ | $1.7 \pm 2.1$ | $1.0 \pm 0.0$ | $1.1 \pm 1.7$ |

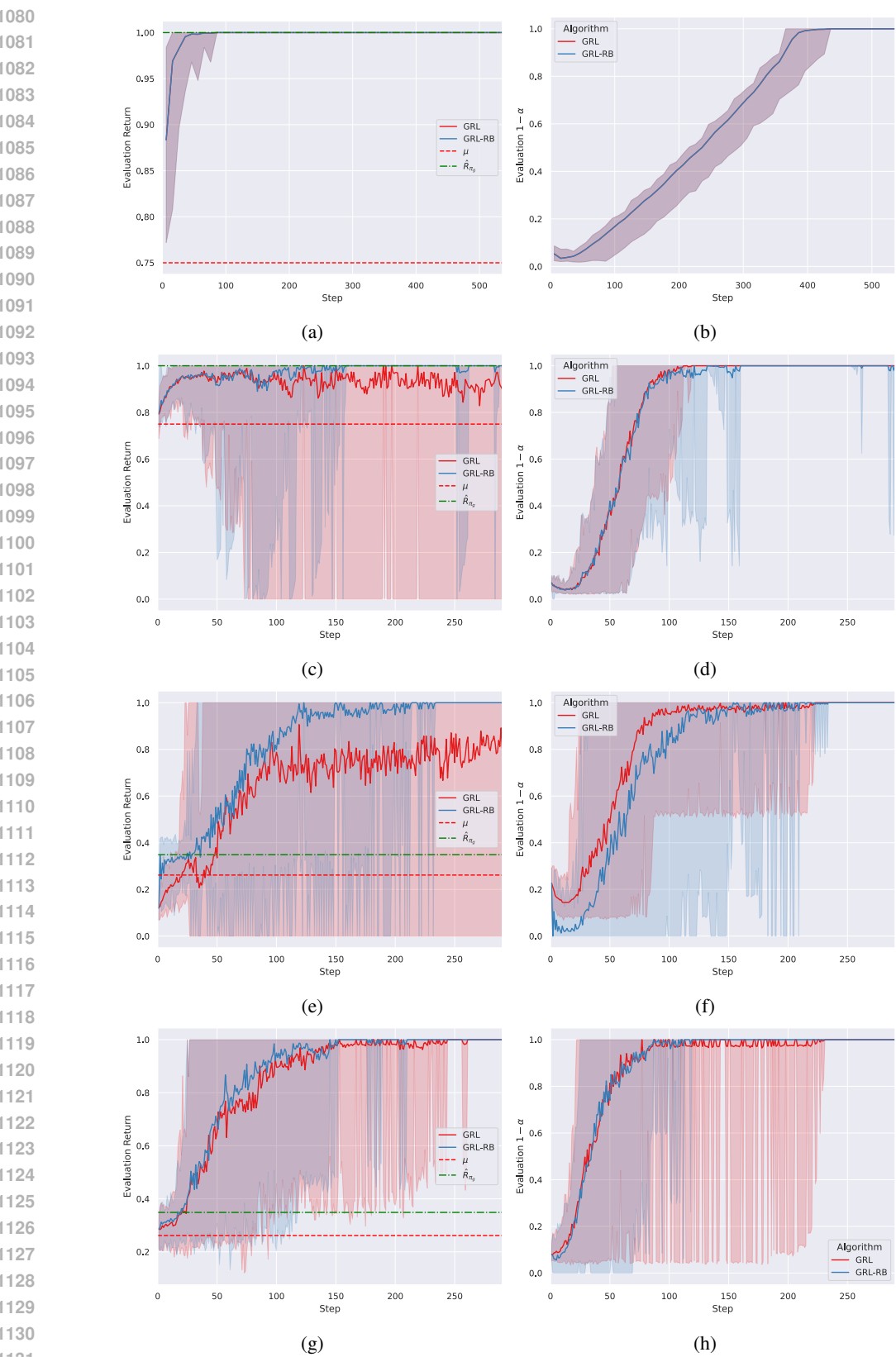

Figure 8: Results for GRL and GRL-RB in the Combination Lock environment. Shading shows minimum and maximum scores across 50 random seeds. a) Optimal guide, terminating learner, good parameter choices. c) Optimal guide, terminating learner, poor parameter choices. e) Non-optimal guide, terminating learner. Poor parameter choices. g) Non-optimal guide, non-terminating learner, poor parameter choices. b,d,f,h) Sampling rates for a,c,e,g) respectively.

### G.1.2 ANTMAZE

Since an implementation of JSRL was not available, we used our own implementation of the algorithm[3]. Default hyperparameters were used for IQL AntMaze experiments, other than those specified in Table 3. For the offline pre-training component, we used the dataset with 1 million transitions. Due to noted issues with the high variance of AntMaze-v0 environments[4], the updated AntMaze-v2 environments were used for these experiments. As such, we highlight that the results for JSRL and IQL may not match those from the corresponding papers. Since the v2 environments address the evaluation instability issue, 4 random seeds (0-4) were used for each AntMaze experiment. Furthermore, we decided not to include the offline dataset in the replay buffer for these experiments, to demonstrate that this mechanism functions even for guide policies in the form of rules, where no prior dataset is available. We also do not re-use the parameters of the offline agent (as in the 'warm-start' setting of JSRL). Again, this is to demonstrate that this method works for a prior policy of any form. Finally, we use the 'curriculum' rather than random-switching version of JSRL, as this achieved the best results in Uchendu et al. (2023) for the environments we consider. The JSRL paper did not specify what hyperparameters were used for each AntMaze experiment, so we chose 10 curriculum stages and the middle values of their Table 11 (Uchendu et al., 2023) - a tolerance of 0.05 and a moving average horizon of 5. Since we used 10 curriculum stages for JSRL, we used an initial $1 - \alpha = 0.1$ for GRL-RB. While IQL does not strictly require offline data, the results in Table 6 shows that it struggles without additional offline data in the replay buffer. For fair comparison, the same guide policy (pretrained using offline data) was used for each of IQL, JSRL and GRL-RB.

Table 6: Results of IQL (Kostrikov et al., 2021) after online fine-tuning, when the offline data is added into the replay buffer (left), and when a fresh online buffer is created (right). 1-$\sigma$ uncertainties are shown.

| Environment | Final Return (Offline Data) | Final Return (No Offline Data) |
|---|---|---|
| antmaze-umaze-v2 | **96.75 ± 0.83** | 84.5 ± 8.2 |
| antmaze-umaze-diverse-v2 | 27.5 ± 18.23 | **40.5 ± 29.34** |
| antmaze-medium-play-v2 | **92.5 ± 2.96** | 63.25 ± 13.77 |
| antmaze-medium-diverse-v2 | **90.0 ± 3.94** | 62.25 ± 37.02 |
| antmaze-large-play-v2 | **69.5 ± 6.18** | 27.5 ± 24.27 |
| antmaze-large-diverse-v2 | **60.75 ± 5.21** | 5.5 ± 9.53 |

Table 7: Results for experiments comparing GRL-RB, LD, JSRL and IQL.

| Environment | Algorithm | Final Score |
|---|---|---|
| Antmaze-Umaze-v2 | IQL | **86.8 ± 6.20** |
| Antmaze-Umaze-v2 | JSRL | 82.6 ± 3.41 |
| Antmaze-Umaze-v2 | LD | **89.5 ± 1.71** |
| Antmaze-Umaze-v2 | GRL-RB (Ours) | **90.3 ± 2.71** |
| Antmaze-Medium-Play-v2 | IQL | 65.35 ± 11.9 |
| Antmaze-Medium-Play-v2 | JSRL | 68.7 ± 8.51 |
| Antmaze-Medium-Play-v2 | LD | 83.1 ± 2.25 |
| Antmaze-Medium-Play-v2 | GRL-RB (Ours) | 77.1 ± 8.48 |
| Antmaze-Large-Play-v2 | IQL | 24.5 ± 21.3 |
| Antmaze-Large-Play-v2 | JSRL | 12.1 ± 12.2 |
| Antmaze-Large-Play-v2 | LD | **44.9 ± 7.23** |
| Antmaze-Large-Play-v2 | GRL-RB (Ours) | **55.6 ± 8.77** |

### G.1.3 ANTMAZE ROBUSTNESS EXPERIMENT

This experiment used some challenging hyperparameters to determine relative the robustness of JSRL and GRL-RB, in the most challenging of the AntMaze environments we consider (Antmaze-Large-Play-v2). The number of curriculum stages for JSRL was decreased to 5, and correspondingly the

---

[3]GitHub link to be included in final version.

[4]Described in a comment on line 189: `https://github.com/Farama-Foundation/D4RL/blob/master/d4rl/locomotion/ant.py`

$1 - \alpha$ of GRL-RB was increased to $0.2$. Further, only 20 episodes were used for evaluation, providing a fairly unreliable indication of episode score improvement. These parameters also allow for a faster transfer from $\pi_g$ to $\pi_l$, with GRL-RB showing good stability in this setting. This is shown in Figure 9.

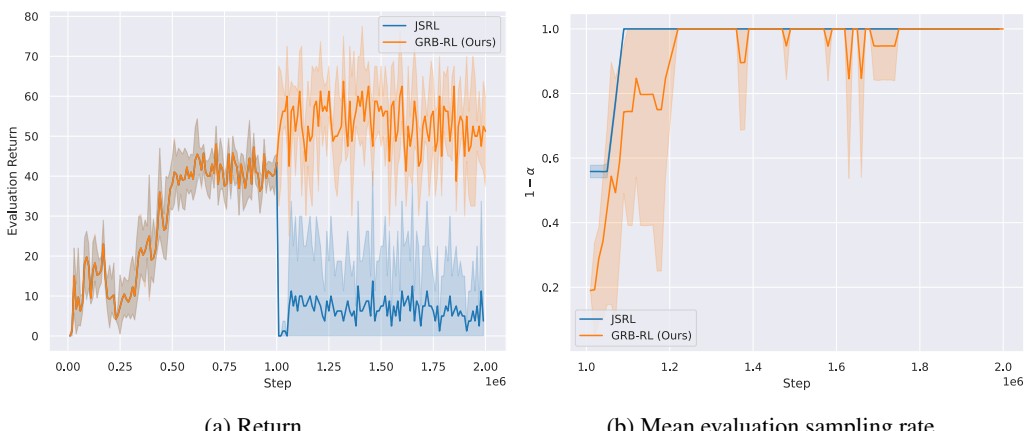

(a) Return.

(b) Mean evaluation sampling rate.

Figure 9: Results for JSRL and GRL-RB for AntMaze-Large-Play-v2 with challenging parameter choices (see text). a) **Return**. Step < 1 million is the offline training phase to show the learning and final return of the guide policy. b) **Mean evaluation sampling rate.** As explain in Section 5, the JSRL method can overuse the learner at the begining of training. GRL-RB rolls back the sample rate when the score starts to drop.

