# OpenReview forum: "Guided Reinforcement Learning with Roll-Back"
_ICLR.cc/2025/Conference — Submitted to ICLR 2025_

### Official Review · Reviewer_UNeW · 2024-10-29

**Soundness:** 2
**Presentation:** 2
**Contribution:** 2
**Rating:** 3
**Confidence:** 3

**Summary:**

The paper addresses guided reinforcement learning, utilizing prior task knowledge to enhance the agent’s learning process. Specifically, it proposes a dynamic sampling rate adjustment for the guide policy, referred to as GRL, along with a variant featuring a roll-back capability, called GRL-RB. The experimental results demonstrate that the proposed methods ensure user-defined performance and outperform other baseline approaches.

**Strengths:**

-  The paper addresses the critical issue of utilizing prior task knowledge to guide the agent's learning, a significant aspect of reinforcement learning.
- The paper provides a comprehensive introduction and overview of related work, laying a solid foundation for the proposed methods.

**Weaknesses:**

Overall, the paper writing and structure needs significant improvement, making it difficult to follow the overall flow.
- The motivation behind the proposed method and its specific details remain largely unclear. For example, in section 3, the authors mention employing a similar sampling approach to Chang et al. (2015) and Chang et al. (2023), as well as a method akin to JSRL (Uchendu et al., 2023). However, it is not clear what these methods entail or how they relate to the current work; readers should not have to refer to external papers for this information. A more detailed formal description of the proposed method should be included in the main body of the text.
- Additionally, several components of the method lack clarity and theoretical justification. For example, the rationale behind using $n_{\pi_l}/t$ as an additional threshold is not explained. Similarly, the reasoning for the new guide sampling rate being $\alpha-(1-\alpha)$ and the theoretical benefits of the roll-back mechanism are unclear. These choices appear to be made heuristically.
- The function of Section 3.2 is also confusing. The derivation of the sampling rate relies on perfect knowledge of the ‘Combination Lock’ task and the guiding policy, which is impractical for practical tasks like the AntMaze task used in this paper. While didactic examples can illustrate theoretical guarantees, the paper lacks more general theoretical results. As it stands, Section 3.2 only suggests that the method works under special conditions that require perfect information.

In summary, while the paper claims to provide a user-defined performance guarantee, it suffers from a lack of clarity and theoretical justification. The assertion that “the assumption of convergence between steps of $\alpha$ is key to the success of GRL” raises concerns. If the "key" to a method relies on an assumption, the authors should reconsider the method and adopt more conservative claims.

**Questions:**

See the weaknesses noted above.

---

> ### Author Response · Authors · 2024-11-21
>
> Thank you for your review and feedback.
>
> Weaknesses
>
> 1. We agree that we could have provided a more detailed explanation of the algorithms, and would be prepared to add this in. The difference if this algorithm’s focus on guaranteeing a certain level of performance when transferred online.
> 2.
>    a. $n\_{\\pi\_l}/t\\%$ is not an additional threshold, it just describes the percentage of time steps during an episode that the learner has been sampled for so far. For example, if we calculate a $\\pi\_l$ sample rate $\\alpha=30\\%$, then we want to ensure the sample rate does not go above 30%. To do this, we simply check that the amount the learner policy has been sampled for is less than $\\alpha=30\\%$, i.e. $n\_{\\pi\_l}/t\\%<30\\%$. It’s just a check to ensure the chosen sampling rate is respected. This is explained in lines 165-167.
>    b. The justification behind $1-(1-\\alpha)$ is this:  if we have $\\alpha\_0 \= 0.9$, giving us $1-\\alpha\_0=0.1$, the first sampling rate will be $\\alpha\_c=\\alpha\_0$. Once convergence has occurred, we assume $\\pi\_l$ has learned to perform equally as well as the guide for that percentage of states, so we can essentially assume that $\\alpha\_c$ now essentially represents 100% use of the guide agent. To meet our performance guarantee, we can only then decrease this by the originally derived $1-\\alpha\_0=0.1$. This is explained in lines 199-202.
>    c. We are not entirely sure what you mean by the theoretical benefits of the roll-back mechanism, the justification is that if the performance falls below the performance threshold, then the sampling of the learner agent is rolled back to the sample rate that achieved the previous best performance, thus giving the agent the opportunity to continue learning at this ‘easier’ stage, where the guide is used more often.
> 3. We agree to the extent we discuss the limitations in the discussion, however the reward schemes we derived sampling rates for are reasonably common in RL, and we provide these theoretical results as a starting point for ourselves and others to further develop methods to guarantee a level of performance during the shift to online learning. We accept there can be convergence challenges, and we introduced the roll-back mechanism to help with that (to relax the assumption you mention in the sentence in your summary, “the assumption of convergence between steps of $\\alpha$..”), but we believe other researchers may find these derivations useful in developing their own methods of meeting the convergence challenge.

---

### Official Review · Reviewer_PsrW · 2024-10-29

**Soundness:** 3
**Presentation:** 2
**Contribution:** 2
**Rating:** 3
**Confidence:** 3

**Summary:**

The authors identify issues in prior works dealing with guided reinforcement learning and aim to address them by deriving an adaptive guide policy sampling rate, which should enable fast & stable transfer of the guide knowledge to the learner policy, while at the same time maintaining performance above a defined maximum tolerable degradation threshold. The method is implemented on top of the commonly known IQL algorithm.

**Strengths:**

I like the general idea of the paper & the derivation of guarantees under some (strong) assumptions are interesting on their own. To the best of my knowledge those are novel & could be helpful in some (limited) scenarios.

**Weaknesses:**

Main weaknesses from my point of view:

1) The empirical evaluation is not very comprehensive:
There are 2 experiments in the CombinationLock environment, which are nice to get an intuitive understanding of the method, however are not very realistic RL tasks. Then there are 3 experiments on Antmaze, in which the LD baseline performs similarly well as the new algorithm. As the authors motivate the method from a quite applied perspective, I am wondering whether that means that there is not much expected gain from the method in practical scenarios. Also, I believe the positive dense reward setting is derived, but not experimented with.

2) The assumptions leading to the derived sampling rates appear to be violated:
The authors realise that in practice the assumptions they make for the derivations are violated & thus add the Roll-Back method in order to improve performance. While the derived results are of course interesting on their own, one has to ask from a practical standpoint whether deriving & using something based on wrong assumptions makes a lot of sense - especially when taking into account the not so convincing empirical evaluation.

3) Slightly overstated claims:
E.g. in lines 67-69, you talk about "a guided RL approach, [...] with a guaranteed online performance above a user-defined threshold". This sounds like a broad claim which I think needs to be qualified further (only mean performance is above threshold, individual episodes can be below; limited to certain reward scenarios, i.e. not for general MDP / RL; only under questionable assumptions, i.e. convergence). The authors also still talk about guarantees when the Roll-Back approach is added in ("A GRL with a roll-back algorithm (GRL-RB) that helps to retain the performance guarantee of GRL while relaxing its assumptions"), even though at that point it becomes clear that there is no guarantee (Roll-Back happens exactly when guarantee is violated; plots show performance can be much below threshold for a long time). Generally I think one has to be more cautious with the term guarantee, e.g. in the abstract it is formulated better.

4) Limited applicability when compared to all possible reward landscapes:
This is not such a big issue in general since guarantees even in just a few environments would be helpful. It's also briefly mentioned by the authors in the discussion.

5) I think Fig 4a might be misleading - it's not clear when something is rolled back, it just looks like violations occur in almost every step.

It may be that I am misunderstanding some issues, so I am prepared to raise my score if the points can be addressed.

**Questions:**

- The evaluation sample rates in the bottom plots 1-3 don't align with what I expected, i.e. why are the percentages of the static baselines not exactly .25 & .75 (e.g. fig 2b has them below .2 & at .6)? I presume the variations (shaded region) of the baselines are the difference between true parameter and sampling? The LD baseline does not appear to really reduce alpha by the same amount every time step, why is that?

- One thing I also don't understand: Why is the alpha not degraded much faster? What I mean is why is it always degraded by $(1-\alpha_0)$ and not by $(1-\alpha_c)$ (algo 1, line 18) - if the assumptions on convergence you make were to hold, the updated alpha should be used for the next degradation or am I missing something?

---

> ### Author Response · Authors · 2024-11-21
>
> Thank you for your time in reviewing our paper and providing feedback.
>
> Weaknesses
>
> 1. The focus of this paper was on deriving the expressions for the sampling rate, and introducing the roll-back mechanism for when it is difficult to ensure convergence. This used most of the available space, so we included enough experiments to demonstrate the results met our theoretical expectations. While LD has a similar design to our algorithms, it does not come with the performance guarantee and issues with it being over-conservative are discussed in the text. Figures 1, 2 and 8 have results for the positive reward scenarios.
> 2. Given the decision of the rate at which to sample either the guide or learner policy is one of the most challenging to justify in guided RL methods, we feel the derivations we have provided do give a useful starting point for a choice of sampling rate, despite the issues that can arise with convergence. Our roll-back mechanism is just one way these can be addressed, and we believe our derived sample rate starting points may give other authors (and ourselves, in future work) the opportunity to develop other interesting methods to address this.
> 3. We agree that the language could be improved to better convey that there can be issues with the guarantee caused by lack of convergence. Thank you for providing specific suggestions and examples, that will assist us very much towards improving this issue.
> 4.
> 5. The sampling rate plot (4b) is meant to show when the sampling rate is rolled back, as it shows the changes in GRL-RB sampling rate correspond to the drops in reward. It is difficult to show a 1-to-1 correspondence, however, when the plots are averaged over many runs. Perhaps providing a single run example would show the correspondence more clearly.
>
> Questions
>
> 1. The reason for the rates not exactly adhering to 25/75% is discussed briefly in lines 273-177, however the second reason is perhaps not well explained, so we will try again here. The way we would implement a static sample rate of, for example, 50%, was in each episode, at each time step, to have a 50% chance of choosing the guide. However, since the episode ends as soon as a non-guide (i.e. non-optimal) action is taken, if a learner action is chosen early in the scenario the episode will end and the sampling rate of the guide will be less than expected for that episode. For example, for an episode where a learner action is chosen on the first time step, the actual rate of guide sampling will be 0% for that episode, even if the overall rate is 75%. We reported the average sampling rates achieved per episode, so this means overall the sampling rate can end up being lower than expected. We could have averaged over the aggregated time steps from all episodes, which would have averaged out to the correct rate, but we felt our averaging method was more correct for an episodic environment. Similarly for the LD method there can be some variations in the actual amount reduced per time step depending on when the episode ends.
> 2. What we are trying to show is this: e.g. if we have $\\alpha\_0 \= 0.9$, giving us $1-\\alpha\_0=0.1$, the first sampling rate will be $\\alpha\_c=\\alpha\_0$. Once convergence has occurred, we assume $\\pi\_l$ has learned to perform equally as well as the guide for that percentage of states, so we can essentially assume that $\\alpha\_c$ now essentially represents 100% use of the guide agent. To meet our performance guarantee, we can only then decrease this by the originally derived $1-\\alpha\_0=0.1$. So, at each successful evaluation, we decrease $\\alpha_c$ by 0.1, giving us successive $\\alpha_c$s of \[0.9, 0.8, 0.7, 0.6, 0.5, 0.4, 0.3, 0.2, 0.1 ,0\].

---

### Official Review · Reviewer_F1aN · 2024-10-29

**Soundness:** 2
**Presentation:** 2
**Contribution:** 1
**Rating:** 3
**Confidence:** 4

**Summary:**

The paper introduces a sampling method that alternates actions between a guide policy (which can originate from various sources, such as model-based or imitation learning methods) and an online learning policy. The main contribution appears to be a mechanism that balances data sampling between the guide and online learning policies by adjusting a user-defined sampling rate, denoted as $\alpha$, which is based on policy performance.

**Strengths:**

The method effectively alternates actions from expert and learning policies, enabling the learning policy to leverage its own actions, a crucial factor for self-correcting estimations. Additionally, it is notable that GRB-RL appears resilient to distribution shift following pre-training, though more extensive testing is needed to confirm this observation.

**Weaknesses:**

* *Lack of novelty*: The primary drawback is limited novelty, as similar approaches that interleave actions from expert and learning policies have been previously explored. The paper acknowledges this by referencing related works and including them as baseline comparisons (e.g., [1]).

* *Limited testing and baseline comparisons*: The approach was evaluated only within variants of the AntMaze environment considering high-dimensional state space environments, which limits insights into its broader applicability. Testing in more diverse environments, such as those found in Gymnasium, like Atari games or other complex benchmarks, could reveal the method's adaptability to different reward structures and exploration needs. Additionally, while the paper reviews a broad range of related works, the baseline algorithms included tend to underperform compared to simpler approaches, such as linear decay (LD) sampling. Also, it would also be interesting to assess how GRL-RB performs with varying data budgets during pre-training.

* *User-defined sampling rate*: The proposed method requires a user-defined sampling rate $\alpha$, which determines the proportion of actions sourced from the expert or the learning policy. While this flexibility can accelerate learning, it places a significant burden on the user to fine-tune $\alpha$, which may hinder practical applicability. Prior work, such as [2], explores this trade-off and provides insights into the optimal balance of data from both sources (expert and learner) to mitigate overestimation issues. Drawing from these insights might further constrain and optimize $\alpha$'s range.

**Overall Assessment**

Overall, the paper lacks novelty, a thorough baseline comparison, and diverse environment testing. The structure is sometimes difficult to follow, and I often found it necessary to consult the Appendix to clarify the role of certain variables. For future improvements, I recommend decoupling the method from its reliance on specific test environments and applying it to more complex, generalizable tasks.

**General remarks**

-Section 2. “Guide, Learning …”: Can be confusing how a policy $\pi$ is derived from an offline and online policy, since either actions are sampled from one of the previous ones. I suggest considering only policies $g$ and $l$ in the notation.

-Line 201 “the new guide sampling rate…” the authors should distinguish between the current $\alpha$ and the initial one used to update the former.

-The authors claim throughout the text that some features, such as the “roll back,” can, for instance, “speed up the transfer learning” before showing the results or evidence to support this (see section 4). Some (important) results are mentioned in the paper but are available in the Appendix.

-I think the paper would benefit from having a dedicated section to describe the baselines IQL and JSLR, and eventually any other method that could be included for comparison.

**References**

[1] Ikechukwu Uchendu, Ted Xiao, Yao Lu, Banghua Zhu, Mengyuan Yan, Joséphine Simon, Matthew Bennice, Chuyuan Fu, Cong Ma, Jiantao Jiao, Sergey Levine, and Karol Hausman. 2023. Jump-start reinforcement learning. In Proceedings of the 40th International Conference on Machine Learning (ICML'23), Vol. 202. JMLR.org, Article 1439, 34556–34583.

[2] Ostrovski, Georg, Pablo Samuel Castro, and Will Dabney. "The difficulty of passive learning in deep reinforcement learning." Advances in Neural Information Processing Systems 34 (2021): 23283-23295.

**Questions:**

[Q1] The authors show in section 3.2.3 a distinction between negative and positive dense rewards. What about a reward normalization such as [-1, 1]?

[Q2] I wonder if tested in more environments, eventually $\alpha$ would get stuck and not converge towards 0, even employing the roll back mechanism. Would that possibly happen?

[Q3] Have you tried to contact the authors of JSLR to obtain its implementation? It seems that their main result is a combination of IQL + JSLR. Is this the way it was tested in this work?

[Q4] Why not consider different amounts of data during the pre-training phase?

[Q5] In G.1.2, could you clarify why you haven't employed the same parameters of the offline agent?

---

> ### Author Response · Authors · 2024-11-21
>
> Thank you for your time in providing us with this detailed feedback.
>
> Weaknesses
>
> 1.  We agree that many guided policy methods exist, however we believe our focus on providing a performance guarantee during the shift to online is our novel contribution.
> 2. The focus of this paper was on deriving the expressions for the sampling rate, and introducing the roll-back mechanism for when it is difficult to ensure convergence. This used most of the available space, so we included enough experiments to demonstrate the results met our theoretical expectations. We agree that experiments with differing data budgets could be included in future work, however given the comparison baselines were all provided the same initialisation, we still believe the provided experiments are sufficient in showing GRL-RB’s effectiveness.
> 3. The purpose of deriving equations for $\\alpha$ for different reward structures is so that the user does not have to finetune $\\alpha$. For the reward structures considered, it allows the user to calculate $\\alpha$ exactly. All the user must do is define the *performance threshold* $\\mu$ they wish to maintain (e.g. 75% of the performance of the guide).
>
> General Remarks
> We thank you for your suggestions, and agree that they would improve the clarity of the paper.
>
> Questions
>
> 1. The initial derivations for $\\alpha$ were only completed for some common reward structures, and interleaved positive/negative reward structures were not included in this work. This limitation was mentioned in the discussion section as an area for future work (lines 467-469).
> 2. It should not be possible as long as the algorithm chosen for learning online is capable of learning. If the algorithm is so poor that it is not capable of learning, then $\\alpha$ would not be reduced. However, this would be the correct behaviour in order to maintain performance above the user defined threshold. We discuss our requirement that the online algorithm be sufficient for learning on line 198, Section 3.1.1.
> 3. Given that JSRL was a relatively straightforward procedure on top of IQL, we decided to write our own implementation, so we did not contact the authors. We also used IQL as the base algorithm, as discussed on line 209, Section 3.1.2.
> 4. We felt this would be an unnecessary comparison, especially as we provided several experiments in Combination Lock and the robustness experiment in AntMaze which shows guides at different levels of capability.
> 5. Thank you for pointing out the unclear wording here \- we should have stated that “we do not re-use *learned* parameters (weights and biases) of the offline agent”. As stated on line 1145, this is to demonstrate that this method works for prior policies of any form, including those that do not have offline policies with weights and biases (such as a set of rules).

---

### Official Review · Reviewer_SUJU · 2024-10-31

**Soundness:** 2
**Presentation:** 2
**Contribution:** 1
**Rating:** 3
**Confidence:** 3

**Summary:**

This paper presents a guided reinforcement learning method, GRL-RB, which adaptively balances the use of the guided policy and RL agent, providing the first performance guarantee in guided RL.

**Strengths:**

1. The main strength is that the paper provides a theoretical view on the selection rate of the expert policy in guided RL setup, which is important and novel.

**Weaknesses:**

1. The major weakness is the lack of existing baselines for this problem. For example, [1, 2] both focus on adaptively learning when to query the experts.
2. The presentation can be improved.
   * Line 79, the space also relates to the time step T;
   * Line 189, the definition of 'Combination Lock MDP' can be further explained with formulas;
   * For Figure 1,2, the same legend should use the same colours. In Figure 3, there is no legend for the expert performance.
   * The derivations on $\alpha$ should be explained more, regarding where it comes from and what it means.
3. The problem seems like an on-policy/off-policy problem, and it is not clear why an offline RL baseline is considered.

[1] Biedenkapp, André, et al. "TempoRL: Learning when to act." *International Conference on Machine Learning*. PMLR, 2021.
[2] Schulz, Felix, et al. "Learning When to Trust the Expert for Guided Exploration in RL." *ICML 2024 Workshop: Foundations of Reinforcement Learning and Control--Connections and Perspectives*.

**Questions:**

See weakness.

---

> ### Author Response · Authors · 2024-11-21
>
> Thank you for your time and effort in reviewing our paper.
>
> Weaknesses
>
> 1. Thank you for your comments. As stated in this paper, to the best of our knowledge this is the first guided policy method that has been produced to focus on guaranteeing a level of performance when transferring from off- to online learning, so while there are not a large number of existing baselines, we used the IQL and JSRL baselines as examples of state-of-the-art approaches to this problem that are closest to our approach. In your point 3\. below, we believe you are referring to IQL as an offline baseline, however in the IQL paper \[1\] Section 5.3, it describes how it can be used for online fine-tuning. For the references you have included, \[2\] does have a similar aim to ours, however their focus on deciding how many time steps for which to use the expert guidance is very similar to the JSRL approach, which we already included as a baseline. We could have included this as a reference, however. \[1\] does not seem to make use of expert guidance, apologies if we have missed something.
> 2.
>      a. Would you mind please elaborating on the issue on Line 79?
>      b. Appendix C provides a full description of the environment, however the explanation in the text could be summarised with equations.
>      c. Thank you for noticing these Figure errors. We will correct them.
>      d. The full derivations were provided in Appendix E to save space.
> 3. See answer to point 1\.
>
> \[1\] Kostrikov, Ilya, Ashvin Nair, and Sergey Levine. "Offline Reinforcement Learning with Implicit Q-Learning." *Deep RL Workshop NeurIPS 2021*.

---

### Meta-Review · Area_Chair_B7fz · 2024-12-21

**Metareview:**

The paper studied guided reinforcement learning, which introduces a sampling method that alternates actions between a guide policy and an online learning policy. Specifically, it proposes a dynamic sampling rate adjustment for the guide policy, referred to as GRL, along with a variant featuring a roll-back capability, called GRL-RB.

Strengths:
The experimental results demonstrate that the proposed methods ensure user-defined performance and outperform other baseline approaches, and also provide a theoretical view on the selection rate of the expert policy in guided RL setup.

Weaknesses:
There are several major weaknesses raised by the reviewers, including:
1. Lack of testing environment and existing baselines, as pointed out by Reviewer SUJU, Reviewer F1aN, and Reviewer PsrW.
2. Lack of novelty, as pointed out by Reviewer F1aN.
3. Writing and presentation, as pointed out by Reviewer SUJU, Reviewer PsrW, and Reviewer UNeW.

All reviewers voted to reject the paper, and it appears that the current version of this paper is not ready to be published in ICLR.

**Additional Comments On Reviewer Discussion:**

None of the reviewers were convinced and changed their minds during the rebuttal period.

---

### Decision · Program_Chairs · 2025-01-22

Reject